# Exploring Attitudinal Dimensions of Inclusive Education: Predictive Factors among Romanian Teachers

Andra Maria Jurca [1], Damaris Baciu [1,2,*], Anca Lustrea [3,*], Simona Sava [3,*] and Claudia Vasilica Borca [3,*]

1 Doctoral School of Psychology-Educational Sciences, West University of Timisoara, 300223 Timisoara, Romania; andra.jurca@e-uvt.ro

2 Doctoral School Programme in Educational Sciences, University of Torino, 10124 Torino, Italy

3 Department of Educational Sciences, University Clinic of Therapies and Psycho-Pedagogical Counseling, West University of Timisoara, 300223 Timisoara, Romania

\* Correspondence: damaris.pungila98@e-uvt.ro (D.B.); anca.lustrea@e-uvt.ro (A.L.); lidia.sava@e-uvt.ro (S.S.); claudia.borca@e-uvt.ro (C.V.B.)

**Abstract:** Inclusive attitudes are considered an important predictor of the quality of educational inclusion. Child-related, teacher-related, and environment-related factors were measured over time in connection with teachers' positive inclusive attitudes. This study aimed to contribute with insights from Romania to the comprehensive understanding of the attitudinal dimensions of inclusive education and the factors that predict it. A quantitative, non-experimental, correlational research design was undertaken in September–October 2022 to determine the factors that can significantly predict the dimensions of inclusive attitudes. A convenience sample of 1040 Romanian teachers participated in the study. The MATIES scale was used to measure the dimensions of inclusive attitudes: cognitive, affective, and behavioral. The results showed that there are a number of universally known factors that have been found to predict inclusive attitudes, like the school environment, close relationships with people with disabilities, and training in special education. Their predictive power is relatively low, ranging between 2% and 9%, suggesting the presence of unexplored influential variables and emphasizing the need for future studies to consider additional factors. The specific and significant factor for Romanian culture was found to be the need for training in special education. The data can be informative for curriculum designers, training providers, and policymakers, signaling the need for comprehensive training in special education in the initial and continuous training of all teachers.

**Keywords:** inclusive education; attitudes; predictive factors; Romanian teachers; MATIES scale

## 1. Introduction

Inclusive education (IE) was the focus of the 1994 Salamanca World Conference on Special Education, which recommended that educational systems around the world should include all students.

Special education involves providing tailored instructional strategies and support to students with diverse learning needs or disabilities. Inclusive education, on the other hand, aims to create a learning environment where all students, including those with disabilities, are educated together in mainstream classrooms, fostering diversity and equal opportunities.

IE was defined as a pedagogical approach that provides equal educational opportunities for all students, regardless of their abilities, disabilities, backgrounds, or other characteristics. At its core is the belief that all learners have the right to quality education and should be actively involved in all aspects of the learning process [1]. Since then, the meaning of inclusive education has evolved from a school prepared and open to all [2] to the celebration of classrooms with diverse students [3].





There has been a lot of research conducted on inclusion concerns. Concerns about the inclusion process were identified as relating to the classroom (class size, behavior management, negative attitudes), the instructional process (curricular design, meeting individual needs, assessment), the school (teacher preparedness, support services, infrastructure, financial resources), and the teacher (competence in teaching in inclusive classrooms, self-confidence, self-esteem, training in inclusive educational practices) [4].

Although teachers' positive attitudes do not directly result in implementing inclusive practices [5], teachers' attitudes have a big impact on the implementation of inclusion, whether at the classroom or school level [6]. To offer quality inclusive services, all future teachers should be trained for inclusive education [7]. Initial training programs should aim to develop inclusive teaching competencies. In parallel, continuous development training programs should be proposed to keep them up to date and diversify their didactic methodology, especially for teachers from specializations other than special education [8].

Given the extensive amount of research in this area, it is clear that attitudes toward inclusive education have been a focus of research for many years [9]. However, the analysis of the literature by Guillemot, Lacroix, and Nocus [10] suggests that more studies are needed in less developed countries in order to generalize the findings on the impact of various factors on teachers' inclusive attitudes. In addressing this need for further research, our study makes a distinctive contribution by providing new evidence from the Romanian cultural context. Specifically, we aim to enhance the understanding of the attitudinal dimensions of inclusive education and identify the factors predicting them. This unique focus on the Romanian cultural context fills a gap in the literature, offering insights that contribute to the broader discourse on inclusive education. The MATIES scale, assessing teachers' inclusive attitudes, was administered to 1040 Romanian teachers. Utilizing a quantitative, non-experimental, correlational research design, the study determined the predictive power of various factors on inclusive attitudes.

### 1.1. The Context of Inclusive Education in Romania

The history of inclusive education in Romania evolved from the early stages of the first pro-inclusion law in 1924, which stipulated the creation of special classes in mainstream schools [11] to the current situation with an ascending trend towards inclusion [12]. In Romania, inclusive education evolved through distinct phases: during the communist era, a segregationist system prevailed with only special schools [13]. The post-communist transition revealed flaws in the educational system, especially concerning children with disabilities, who were often institutionalized due to a lack of specialized teachers. The transition period involved stages of deconstruction, restructuring, and comprehensive educational reform, marked by collaborative efforts, project promotions, and the development of materials in partnership with national and international organizations [14]. Post-2007, as a full member of the European Union, Romania embraced pro-inclusive policies, implementing a National Strategy and corresponding Action Plan. A positive trend emerged, with an increasing number of students with disabilities being included in mainstream education.

The number of children with disabilities included in Romania [12], but also at the international level [15], is constantly increasing. Educational systems regularly deal with an increased number of diverse students, but without being fully prepared. Currently, it is estimated that more than 50% of students with disabilities are included [16]. But, progress towards that goal has been slow, and one of the barriers identified is teachers' attitudes towards inclusive education [17].

The findings of Romanian studies on IE show a spectrum of attitudes, from institutional violence and stigma [18] to a predominantly favorable attitude toward inclusion [19,20]. Among the factors that influence attitudes are the level of professional qualification, experience in the field of special education [20], nature and severity of the disability [21,22], or perceived self-efficacy [23].

*1.2. Teachers' Attitudes towards Inclusive Education*

An understanding of multidimensional attitudes is recommended for measuring teachers' attitudes towards inclusive education [24]. Attitudes include three essential components: the affective component, which is responsible for the emotional aspect; the cognitive component, which provides the basic beliefs and arguments; and the behavioral component, which is how the attitude is intended to be translated into action. Eagly and Chaiken's composite attitude–behavior model highlights the complexity and multi-dimensionality of attitudes, emphasizing the interdependent relationship between emotions, beliefs, and actions that shapes an individual's overall attitude towards a particular issue or object [25]. Another explanatory theory could be the Knowledge–Attitude–Behavior (KAB) theory, which posits a sequential relationship between knowledge, attitudes, and behavior, suggesting that acquiring knowledge influences attitudes, which in turn shape behavior. Applied to our research on inclusive attitudes, this theory guides our investigation by recognizing that increasing knowledge about inclusive education can positively influence educators' attitudes and subsequently lead to more inclusive practices in the classroom. The theory of planned behavior (TPB) [26] has been used by many attitudinal studies [27]. The TPB asserts that individual behavior is influenced by three key factors: attitudes, subjective norms, and perceived behavioral control. In the context of our research on inclusive attitudes, this theory serves as a guiding framework. We use it to comprehend how subjective norms and perceived behavioral control shape teachers' attitudes toward inclusive education, which in turn shape their actual behaviors and intentions in fostering inclusive practices.

The research findings on the factors that impact teachers' inclusive attitudes are diverse; results reported both positive and, on the contrary, negative impacts of the same factor in various cultures [28]. Only the type of school predicted the same result across countries: literature reviews have shown that special education teachers tend to have more favorable attitudes toward IE than regular education teachers [10,29–31].

Several factors influencing teachers' inclusive attitudes were considered: age [19], rural or urban work environment [32–36], teaching experience [31], experience with students with disabilities [31], training in inclusive education [37], the perceived need for special education training [38], and having a family member with a disability [30,35,37,39]. Younger teachers exhibited more positive attitudes, as did those in rural settings, with fewer years of teaching experience and prior experience working with students with disabilities. Additionally, teachers who participated in inclusive education training expressed a need for special education training, and those who had a family member with a disability also demonstrated more positive attitudes [29,31].

The impact of these factors on the three attitudinal dimensions has been analyzed in various studies. Dignath et al. [40] conducted a meta-analysis on the impact of several factors on beliefs (cognitive dimension). The factor with the greatest impact on teachers' beliefs about inclusion was the training in special education. The affective dimension was analyzed, especially in light of the previous experience with people with disabilities [30]. Having a family member with a disability has a positive impact on sentiments towards IE. The behavioral dimension (the intention to act) was shown to be influenced mainly by the teaching experience [41].

In the Romanian context, the factors measured in relation to the teachers' inclusive attitudes were the type of school (teachers from special education had more positive attitudes than those teaching in mainstream education), the school environment (those from rural education had more positive attitudes than those from urban schools [42]), and age and professional experience were also studied, without significant correlations [43]. However, the studies that have been carried out so far for the Romanian context have taken into account only a few factors and have not come to any definitive conclusions about the factors that influence the attitudes of the teachers towards inclusive education. Due to the diversity of the reported results on inclusive attitudes, this study aimed to determine the factors that significantly predict the dimensions of inclusive attitudes in Romania and to provide new evidence from the Romanian cultural context.

## 2. Methodology

### 2.1. Research Design

To answer the research question: What are the factors that significantly predict teachers' inclusive attitudes (their beliefs, sentiments, and behaviors) toward inclusive education?, a quantitative, non-experimental, correlational research design was conducted. The aim of this study is to identify the factors that significantly predict teachers' attitudes on the three dimensions (beliefs, sentiments, and behaviors) towards inclusive education in Romania.

The predictive factors for the attitudinal dimensions taken into consideration were: teachers' age, teaching experience, school environment (urban/rural), school type (mainstream/special school), the teachers' experience in teaching students with disabilities, having a student with a disability in the classroom, initial training in special education, continuing training in special education, the need for training in special education, and the presence of a person with a disability in the family.

The following hypotheses were advanced:

1. The cognitive dimension of inclusive attitudes (the beliefs about inclusion) is best predicted by the teachers' training in special education.

   This hypothesis builds on the idea that acquiring knowledge through special education training positively influences teachers' beliefs, subsequently shaping their attitudes and potentially leading to more inclusive behaviors in line with the KAB framework.

2. The affective dimension of inclusive attitudes (sentiments about inclusive education) is best predicted by the existence of a significant relationship with people with disabilities.

   This hypothesis is grounded in the belief that personal connections with individuals with disabilities can profoundly impact teachers' emotional responses, fostering positive affective attitudes toward inclusive practices [44].

3. The behavioral dimension of inclusive attitudes (intention to implement inclusive practices) is best predicted by the teachers' experience in teaching students with disabilities.

   This assumption aligns with the TPB [26], suggesting that direct experience influences behavioral intentions and the likelihood of implementing inclusive practices.

4. The teachers' positive attitudes toward inclusive education are best predicted by their training in special education.

   In line with the Knowledge–Attitude–Behavior (KAB) theory suggesting that acquiring knowledge through specialized training positively influences attitudes and subsequently shapes teachers' overall positive disposition toward inclusive education, the 4th hypothesis was formulated. The theory posits a sequential relationship where knowledge acquisition leads to favorable attitudes, ultimately impacting behavior and emphasizing the pivotal role of training in shaping educators' attitudes.

### 2.2. Data Collection

The data were collected as part of the research project funded by the Faculty of Sociology and Psychology of the West University of Timisoara. An online questionnaire was run through the Google Forms platform from September to October 2022. The online questionnaire was distributed through school inspectorates across the country. The data collection method used resulted in a convenience sample, obtained via a snowball sampling, which does not allow the response rate of participants to be estimated.

### 2.3. Participants

A total of 1040 teachers from all Romanian counties participated in this study from different school environments, types of education, and specializations (see Table 1 for a detailed presentation of the sample). Their ages ranged from 19 to 68 years (M = 43.36, SD = 9.91). Their teaching experience ranged from 0 to 46 years (M = 17.84, SD = 10.85). Of the participants, 115 (11%) were principals, 6 (0.6%) inspectors, 17 (1.6%) school counselors,

35 (3.4%) support teachers, 140 (13.5%) preschool teachers, 216 (20.8%) primary teachers, and 511 (49.1%) secondary/high school teachers. There were also 359 (34.5%) teachers teaching in rural areas and 681 (65.5%) in urban areas. The teachers who participated in this study work in both special education 208 (20%) and mainstream education 832 (80%).

**Table 1.** Descriptive summary of the sample.

| Characteristics | N | % |
|---|---|---|
| School environment | | |
| Rural | 359 | 34.5 |
| Urban | 681 | 65.5 |
| School type | | |
| General and Mainstream schools | 832 | 80 |
| Special schools | 208 | 20 |
| Had taught students with disabilities | | |
| yes | 928 | 89.2 |
| no | 112 | 10.8 |
| Special education courses in initial training | | |
| 0 | 530 | 51 |
| 1 | 233 | 22.4 |
| 2 | 92 | 8.8 |
| 3 or more | 185 | 17.8 |
| Special education courses in continuous training | | |
| 0 | 324 | 31.2 |
| 1 | 267 | 25.7 |
| 2 | 146 | 14 |
| 3 or more | 303 | 29.1 |
| Need for training in special education | | |
| yes | 792 | 76.2 |
| no | 248 | 23.8 |
| Presence of a family member with disabilities | | |
| yes | 792 | 76.2 |
| no | 248 | 23.8 |
| Has a student with disabilities in the classroom | | |
| yes | 727 | 69.9 |
| no | 312 | 31.1 |
| Experience in teaching students with disabilities | | |
| yes | 928 | 89.2 |
| no | 112 | 10.8 |

*2.4. Instrument*

The Multidimensional Attitudes towards Inclusive Education Scale (MATIES) [45] was used to measure the dimensions of inclusive attitudes. We chose to use this scale because it is one of the most widely used measurement tools due to its good psychometric properties and investigative focus [24]. The MATIES measures teachers' attitudes towards inclusive education in three dimensions: cognitive (perception, beliefs), affective (sentiments), and behavioral (intention). Each of these dimensions had good internal consistency, above the 0.7 value for all dimensions and for the whole scale. In order to collect the data, the retroverse translation technique was used from English into Romanian and back, and the wording of the items was adapted to correspond to the cultural context of the country. After direct and back-translation, the opinion of three experts, university professors, was sought to ensure cultural appropriateness. The Romanian version of the instrument has a good internal consistency of 0.87.

The MATIES scale consists of 18 items, and responses are measured on a 6-step Likert scale, where 1 is strong disagreement and 6 is strong agreement. Each dimension covers six items. Items within the affective dimension of attitudes address teachers' feelings and emotions associated with inclusive education, formulated like: I am pleased that students with a disability are able to attend the local neighborhood school. Items within the cognitive

dimension of attitudes reflect teachers' perceptions and beliefs about inclusive education (for example: I believe that all students should be able to study side by side in the regular classroom regardless of ability). Finally, items such as I am willing to modify the physical environment to include students with a disability in the regular classroom are statements of behavioral intent and imply the teacher's intention to act in a certain manner toward inclusive education.

Also, a demographic survey was added to collect data about the independent variables mentioned above. There were 14 demographic questions that collected data on school environment, type of school, experience in teaching students with disabilities, having a student with a disability in the classroom, special education courses in initial education, special education courses in continuous training, need for training, and the presence of a family member with a disability.

## 3. Results

The results of our study, which aimed to identify the factors that significantly predict teachers' attitudes towards inclusive education in Romania, will be presented according to the three distinct dimensions of inclusive attitudes: cognitive, affective, and behavioral. Additionally, we explore the construct of teachers' overall positive attitudes towards inclusive education. To determine what factors predict positive attitudinal inclusion and its dimensions, a simple multilinear regression was conducted. The results are analyzed below.

### 3.1. Factors Predicting the Cognitive Dimension

To test the first research hypothesis, that the cognitive dimension of inclusive attitudes (the beliefs about inclusion) is best predicted by the teachers' trained in special education, a simple multilinear regression was carried out to identify teachers' attitudes towards inclusive education in the cognitive dimension according to the teachers' age, teaching experience, school environment (urban/rural), school type (mainstream/special school), the teachers' experience in teaching students with disabilities, having a student with a disability in the classroom, initial training in special education, continuing training in special education, the need for training in special education, and the presence of a person with a disability in the family. Regression coefficients and standard errors can be found in Table 2.

As can be seen in Table 2, the multiple regression model statistically significantly predicted the cognitive dimension: F (10,1029) = 11.307, $p < 0.001$, $R^2$ = 0.099. Only the variables of school type, having a student with a disability in the classroom, initial training in special education, continuing training in special education, the need for training in special education, and the presence of a person with a disability in the family were statistically significant to the prediction, $p < 0.05$.

The teachers' beliefs about inclusive education are predicted by the above-mentioned factors in the proportion of 9.9%. The strongest predictor is school type ($\beta$ = −0.28, $p < 0.001$), followed by the need for training in special education ($\beta$ = −0.13, $p < 0.001$), continuing training in special education ($\beta$ = 0.09, $p < 0.05$), initial training in special education ($\beta$ = 0.07, $p < 0.05$), the presence of a person with a disability in the family ($\beta$ = −0.07, $p < 0.05$), and having a student with a disability in the classroom ($\beta$ = 0.06, $p < 0.05$).

While we anticipated teachers' training, particularly in special education, to be the most influential factor in the cognitive dimension, the results contradicted this hypothesis. The most predictive factor is school type, with teachers from mainstream education exhibiting more positive attitudes towards inclusion than those from special education (results supported by [29]).

**Table 2.** Multiple regression results for the cognitive dimension of teachers' inclusive attitudes.

| Cognitive | B | 95% CI for B | | SE B | β | $R^2$ |
|---|---|---|---|---|---|---|
| | | LL | UL | | | |
| Constant | 5.14 *** | 4.52 | 5.77 | 0.32 | | 0.099 *** |
| Age | 0.004 | −0.00 | 0.01 | 0.00 | 0.03 | |
| School environment | 0.072 | −0.07 | 0.21 | 0.07 | 0.03 | |
| Type of school | −0.821 *** | −1.02 | −0.61 | 0.10 | −0.28 *** | |
| Teaching experience | −0.007 | −0.01 | 0.00 | 0.00 | −0.06 | |
| Experience in teaching students with disabilities | −0.222 | −0.46 | −0.01 | 0.12 | −0.06 | |
| Student with a disability in the classroom | 0.169 * | 0.00 | 0.33 | 0.08 | 0.06 * | |
| Initial training | 0.077 * | 0.00 | 0.14 | 0.03 | 0.07 * | |
| Continuing training | 0.090 * | 0.02 | 0.16 | 0.03 | 0.09 * | |
| Need for training | −0.360 *** | −0.51 | −0.20 | 0.08 | −0.13 *** | |
| Presence of a person with a disability in the family | −0.189 * | −0.34 | −0.02 | 0.08 | −0.07 * | |

Note: B = unstandardized regression coefficient; CI = confidence interval; LL = lower limit; UL = upper limit; SE B = standard error of the coefficient; β = standardized coefficient; $R^2$ = coefficient of determination. * Value is significant at $p < 0.05$. *** Value is significant at $p < 0.001$.

### 3.2. Factors Predicting the Affective Dimension

For the second research hypothesis, that the affective dimension of inclusive attitudes (sentiments about inclusive education) is best predicted by the existence of a significant relationship with people with disabilities, a simple multilinear regression was carried out to identify teachers' attitudes towards inclusive education in the affective dimension. The same variables were considered: teachers' age, teaching experience, school environment (urban/rural), school type (mainstream/special school), the teachers' experience in teaching students with disabilities, having a student with a disability in the classroom, initial training in special education, continuing training in special education, the need for training in special education, and the presence of a person with a disability in the family. Regression coefficients and standard errors can be found in Table 3.

**Table 3.** Multiple regression results for the affective dimensions of teachers' inclusive attitudes.

| Affective | B | 95% CI for B | | SE B | β | $R^2$ |
|---|---|---|---|---|---|---|
| | | LL | UL | | | |
| Constant | 4.92 *** | 4.27 | 5.58 | 0.33 | | 0.024 ** |
| Age | −0.005 | −0.01 | 0.00 | 0.00 | −0.04 | |
| School environment | 0.165 * | 0.01 | 0.31 | 0.07 | 0.06 * | |
| Type of school | −0.203 | −0.00 | 0.41 | 0.10 | 0.07 | |
| Teaching experience | 0.000 | −0.01 | 0.01 | 0.00 | −0.00 | |
| Experience in teaching students with disabilities | −0.191 | −0.43 | 0.05 | 0.12 | −0.05 | |
| Student with a disability in the classroom | 0.141 | −0.02 | 0.31 | 0.08 | 0.05 | |
| Initial training | 0.006 | −0.06 | 0.07 | 0.03 | 0.00 | |
| Continuing training | 0.039 | -03 | 0.11 | 0.03 | 0.04 | |
| Need for training | −0.076 | −0.24 | 0.08 | 0.08 | −0.02 | |
| Presence of a person with a disability in the family | −0.221 ** | −0.38 | −0.05 | 0.08 | −0.08 ** | |

Note: B = unstandardized regression coefficient; CI = confidence interval; LL = lower limit; UL = upper limit; SE B = standard error of the coefficient; β = standardized coefficient; $R^2$ = coefficient of determination. * Value is significant at $p < 0.05$. ** Value is significant at $p < 0.01$. *** Value is significant at $p < 0.001$.

The multiple regression model statistically significantly predicted the affective dimension, F (10,1029) = 2.484 $p < 0.01$, $R^2 = 0.024$. Only the variables of school environment

(urban/rural) and the presence of a person with a disability in the family were statistically significant to the prediction, $p < 0.05$.

Teacher's sentiments about inclusive education are predicted in proportion to 2.4%. Also, the strongest predictor is the presence of a person with a disability in the family ($\beta = -0.08$, $p < 0.001$), followed by the school environment (urban/rural) ($\beta = 0.06$, $p < 0.05$). Research hypothesis 2 is validated; we anticipated that having a significant relationship with people with disabilities would exert the strongest predictive influence in the affective dimension, and the results affirmed this hypothesis. Teachers with close relationships with individuals with disabilities exhibit more positive attitudes toward inclusion compared to those without such connections. Similar results were identified in other studies [44].

### 3.3. Factors Predicting the Behavioral Dimension

The third research hypothesis, that the behavioral dimension of inclusive attitudes (intention to implement inclusive practices) is best predicted by the teachers' experience in teaching students with disabilities, was tested using the same simple multilinear regression. To identify teachers' attitudes towards inclusive education in the behavioral dimension, the variables used for the other two dimensions were considered: teachers' age, teaching experience, school environment (urban/rural), school type (mainstream/special school), the teachers' experience in teaching students with disabilities, having a student with a disability in the classroom, initial training in special education, continuing training in special education, the need for training in special education, and the presence of a person with a disability in the family. The data can be observed in Table 4.

**Table 4.** Multiple regression results for the behavioral dimension of teachers' inclusive attitudes.

| Behavioral | B | 95% CI for B | | SE B | $\beta$ | $R^2$ |
|---|---|---|---|---|---|---|
| | | LL | UL | | | |
| Constant | 5.407 *** | 4.73 | 6.08 | 0.34 | | 0.042 *** |
| Age | −0.008 | −0.02 | 0.00 | 0.00 | −0.06 | |
| School environment | 0.120 | −0.03 | 0.27 | 0.08 | 0.04 | |
| Type of school | −0.041 | −0.25 | 0.17 | 0.11 | −0.01 | |
| Teaching experience | −0.002 | −0.01 | 0.01 | 0.00 | −0.01 | |
| Experience in teaching students with disabilities | −0.113 | −0.36 | 0.14 | 0.13 | −0.02 | |
| Student with a disability in the classroom | 0.013 | −0.16 | 0.18 | 0.08 | 0.00 | |
| Initial training | 0.092 * | 0.01 | 0.16 | 0.03 | 0.08 * | |
| Continuing training | 0.058 | −0.01 | 0.13 | 0.03 | 0.05 | |
| Need for training | −0.365 *** | −0.53 | −0.19 | 0.08 | −0.13 *** | |
| Presence of a person with a disability in the family | −0.072 | −0.24 | 0.10 | 0.08 | −0.02 | |

Note: B = unstandardized regression coefficient; CI = confidence interval; LL = lower limit; UL = upper limit; SE B = standard error of the coefficient; $\beta$ = standardized coefficient; $R^2$ = coefficient of determination. * Value is significant at $p < 0.05$. *** Value is significant at $p < 0.001$.

As it can be noticed in the table above, the multiple regression model significantly predicted the behavioral dimension, F (10,1029) = 4.49, $p < 0.001$, $R^2 = 0.042$. Only the variables of initial training in special education and the need for training in special education were statistically significant to the prediction, $p < 0.05$.

As a result of the present data, we can see that teachers' perceptions of inclusive education are predicted in proportion to 4.2%. The strongest predictor is the need for training in special education ($\beta = -0.13$, $p < 0.001$), followed by initial training in special education ($\beta = 0.08$, $p < 0.05$).

A high stated need for training by teachers predicts the existence of positive, real intentions to act towards the inclusion of students with disabilities. The stronger their

intentions to include, the more training they have undergone in special education (results supported by [31]). However, the results do not support research hypothesis 3.

### 3.4. Factors Predicting the Teachers' Inclusive Attitudes

To test research hypothesis 4, according to which 'The teachers' positive attitudes toward inclusive education are best predicted by their training in special education', a simple multilinear regression was carried out to identify teachers' attitudes towards inclusive education. The variables considered were the teachers' age, teaching experience, school environment (urban/rural), school type (mainstream/special school), the teachers' experience in teaching students with disabilities, having a student with a disability in the classroom, initial training in special education, continuing training in special education, the need for training in special education, and the presence of a person with a disability in the family.

The multiple regression model significantly predicted teacher's attitudes towards inclusive education, $F_{(10,1029)} = 5.067$, $p < 0.001$, $R^2 = 0.047$. Only the variables of school type, continuing training in special education, the need for training in special education, and the presence of a person with a disability in the family were statistically significant to the prediction, $p < 0.05$. Regression coefficients and standard errors can be found in Table 5.

**Table 5.** Multiple regression results for teachers' inclusive attitudes.

| Attitude | B | 95% CI for B | | SE B | $\beta$ | $R^2$ |
|---|---|---|---|---|---|---|
| | | LL | UL | | | |
| Constant | 5.16 *** | 4.63 | 5.69 | 0.27 | | 0.047 *** |
| Age | −0.003 | −0.01 | 0.00 | 0.00 | −0.03 | |
| School environment | 0.119 | −0.00 | 0.24 | 0.06 | 0.06 | |
| Type of school | −0.220 * | −0.39 | −0.04 | 0.08 | −0.09 * | |
| Teaching experience | −0.003 | −0.01 | 0.00 | 0.00 | −0.03 | |
| Experience in teaching students with disabilities | −0.175 | −0.37 | −0.02 | 0.10 | −0.05 | |
| Student with a disability in the classroom | 0.108 | −0.02 | 0.24 | 0.07 | 0.05 | |
| Initial training | 0.058 | 0.00 | 0.11 | 0.03 | 0.07 | |
| Continuing training | 0.062 * | 0.00 | 0.12 | 0.03 | 0.08 * | |
| Need for training | −0.267 *** | −0.40 | −0.13 | 0.06 | −0.12 *** | |
| Presence of a person with a disability in the family | −0.160 * | −0.29 | −0.02 | 0.06 | −0.07 * | |

Note: B = unstandardized regression coefficient; CI = confidence interval; LL = lower limit; UL = upper limit; SE B = standard error of the coefficient; $\beta$ = standardized coefficient; $R^2$ = coefficient of determination. * Value is significant at $p < 0.05$. *** Value is significant at $p < 0.001$.

Teachers' attitudes towards inclusive education are predicted in proportion to 4.7% and the strongest predictor is the need for training in special education ($\beta = -0.12$, $p < 0.001$), followed by school type ($\beta = -0.09$, $p < 0.05$), continuing training in special education ($\beta = 0.08$, $p < 0.05$), and the presence of a person with a disability in the family ($\beta = -0.07$, $p < 0.05$). Research hypothesis 4 is infirmed, contrary to our expectations. Specific training does not hold the greatest predictive power over teachers' attitudes. Instead, the expressed need for training emerges as the primary predictor of inclusive attitudes. Teachers who acknowledge their limitations and express a desire to enhance their competence to teach in inclusive classrooms tend to have more positive attitudes toward inclusion.

## 4. Discussion

Inclusive attitudes are considered an important predictor of the quality of educational inclusion [3,46]. The level of investigation of inclusive attitudes is complex, and the literature reviews in the field demonstrate the constant focus on this topic [10,30,31,47]. Many

barriers to effective inclusion were identified in Romania, among them a central element being the teachers' attitudes [48]. This study aimed to contribute to the comprehensive understanding of the attitudinal dimensions of inclusive education and the factors that predict it, offering insights from the Romanian context.

To answer the research question and determine the factors that significantly predict the cognitive, affective, and behavioral dimensions of inclusive attitudes of the Romanian teachers, a quantitative, non-experimental, correlational research design based on the MATIES scale [45] was used, conducting simple multilinear regression analyses.

Firstly, the results showed that the factor that most significantly predicts teachers' inclusive attitudes is their need for training in special education. This concept refers to the teachers' need to further develop skills or competencies that they identify as not having mastered at a desirable level while having the intention to implement IE (the behavioral dimension). The teachers who mostly express their need for training in special education are the most likely to have a positive attitude toward IE, well-articulated beliefs (the cognitive dimension), and the readiness to act towards implementing it (the behavioral dimension). A similar result was displayed by the TALIS study [49], which showed that teachers in Romania expressed one of the most intensive needs for training in teaching students with disabilities. In other EU post-communist countries that share common characteristics, such as a relatively conservative and monolithic culture, as well as similarities in race and religion [50], the identified research conducted in countries like Slovakia [51], Hungary [52], the Czech Republic [53], and Bulgaria [36] reveals a similar high demand for teacher training. The present study is in line with those of DeVault [38], who showed that teachers willing to train in special education have more positive attitudes towards IE. Also, the teachers who recognize their professional development needs have certain distinctive characteristics: openness to change and flexibility [54], self-reflection [55], and lifelong learning attitudes [56]. The results can be explained by the relationship between the openness to change personality trait and the teachers' inclusive attitudes [57]. Another possible interpretation of the need for training as a predictor of the inclusive attitude is that a teacher who identifies their limits and needs and constantly seeks self-improvement is a reflective teacher, focused on continued learning; such a link between reflective practices and openness to inclusion has already been demonstrated [58,59]. Furthermore, studies show that the greater the participation in continuing professional development (CPD) in special education, the more positive the attitudes [31,60], confirming our data related to continued training as an important factor predicting the teachers' attitude towards IE. In light of the Knowledge–Attitude–Behavior (KAB) theory, which outlines a sequential link between knowledge, attitudes, and behavior, enhancing knowledge about inclusive education through training can positively impact educators' attitudes, subsequently influencing more inclusive practices in the classroom.

Secondly, the type of school (mainstream or special school) was revealed to be another predictive factor of inclusive attitudes. Teachers in mainstream schools exhibited a more positive attitude than those teaching in special schools, although other research reported more positive attitudes among teachers in special schools [10,60]. This counterintuitive result can be explained by the fact that in special schools in Romania there are students with profound degrees of disability and with multiple disabilities [61]. Also, the problems of inclusion in Romania are directly experienced by teachers in special education, who work in multidisciplinary teams with specialists in inclusive education. In the context of the TPB [26], the inclination of mainstream teachers towards more positive attitudes might be attributable to perceived behavioral control, as they do not have direct interactions with students with severe disabilities. The lack of direct experience in teaching students with severe disabilities may contribute to a more favorable disposition due to reduced perceived challenges. On the other hand, teachers in special schools, facing the complexities of profound and multiple disabilities, may perceive less control over implementing inclusive practices, potentially shaping less positive attitudes.

Thirdly, another predictive factor is the close relationship with a person with disabilities, a factor found in most studies [30,35,39]. The presence of a person with a disability in the family predicts positive inclusive attitudes; having a close relationship with a person with a disability is associated with more positive attitudes.

The school environment was another factor that predicted teachers' attitudes towards inclusive education on the affective dimension. Teachers in rural areas showed more positive feelings towards the inclusion of students with disabilities, as they usually have a smaller number of students in the classroom. This result is in line with other studies that have identified a more positive attitude among rural teachers [33–36]. One possible explanation may be the specificity of these communities and rural schools. Rural communities have a smaller number of inhabitants, allowing them to know each other better [33,35] and help each other, being open to the community. It is known that an inclusive school is open to the community [62].

Predictive factors found in other studies as well are training, the type of school (mainstream or special school), the school environment (rural or urban), and the relationship with a person with a disability. What is specific to Romania is the need for training in special education. Important factors described in the literature, such as teaching experience or experience in teaching students with disabilities, do not significantly predict the positive inclusive attitudes of teachers in Romania. From a statistical point of view, corroborating these findings, i.e., direct experiential knowledge of the issue, teaching students with disabilities, and experience in teaching students with disabilities, do not significantly predict inclusive attitudes. How can we turn these predictors into positive ones? Through real classroom support, provided by the system, school, and specialists. Similar findings regarding inadequate classroom support are reported in other post-communist countries, such as Hungary [52] and the Czech Republic [53], which highlight the absence of teaching assistants or specialized personnel. Future studies could investigate the extent to which this support is actually provided and the way this support influences teachers' attitudes.

Summing up, this research contributes to international discussions by bringing new evidence of the Romanian context of IE. The factors that seem to influence inclusive attitudes regardless of the cultural context are the type of school, the school environment, the presence in the classroom or in the family of a person with disabilities, and the initial or continuous training in special education. The main factor, specific to Romanian culture, was the expressed need for training in special education and little significance regarding the influence of teaching experience. While the school type is a strong factor influencing beliefs about IE (the cognitive dimension), the presence in the classroom or in the family of a person with disabilities mainly influences the affective dimension. At the same time, the training undertaken by teachers and the need for further training on teaching students with disabilities influence the teachers' intention to implement IE.

These findings draw attention to the need for special education training that Romanian teachers have expressed. Thus, more training courses in special education in initial and continuous teacher training could be provided to future teachers. These courses should focus on developing real competence in teaching in inclusive settings and training reflexive, lifelong learning professionals.

*Limitations*

A first limitation of the study is that some factors were not taken into account. The fact that the analyzed factors (the most frequently mentioned and evaluated in studies) have a low predictive power, between 2% and 9%, shows that there are other important factors that we did not consider in this study. Future studies should take into account factors such as support services provided to teachers, collaboration in multidisciplinary teams, collaboration with parents, or teachers' self-esteem in teaching in IE classrooms.

A possible limitation of this study is the lack of data on the degree of disability. Previous studies grouped the factors into three categories, i.e., child-, teacher-, and school/educational-environment-related [31]. In the present study, we did not take into account

factors related to the child's type and degree of disability. Factors related to this category could better explain why teachers in special schools in Romania have a more negative attitude towards inclusion. Future studies on this topic in Romania could take into account these child-related factors to see if the results are replicated.

Another limitation is our study's reliance solely on quantitative methods. While statistical analyses have provided valuable insights into predictive factors of inclusive attitudes, a purely quantitative approach may overlook nuanced qualitative aspects. Future research could benefit from incorporating qualitative methodologies, such as interviews or focus group discussions, to capture the depth of teachers' experiences and perspectives, enhancing the overall comprehensiveness of the study.

## 5. Conclusions

The primary objective of this research was to identify the predictive factors associated with inclusive attitudes and to contribute to the existing knowledge with evidence from the context of the Romanian educational system. The findings of this study revealed that several universally recognized factors, including the school environment and close relationships with individuals with disabilities, are confirmed to have an influencing power on the teachers' attitude towards IE in Romania, which is similar to the situation in other countries. However, training in special education predicts even stronger inclusive attitudes within the Romanian cultural context. Notably, the expressed need for training in special education emerged as a particularly powerful predictor.

Among the three attitudinal dimensions, cognitive, affective, and behavioral, the behavioral dimension assumes a primary role within the framework of inclusion implementation. It means that not only are one's beliefs or emotions important, but so are the intent and motivation to engage in inclusive practices. The inclination to take action is influenced by two factors: the need for training and the initial training. This implies that, while positive beliefs and emotions are valuable, those individuals who have received initial training, i.e., those who, during their college years, have taken courses that address issues of inclusion or special education, and who express a desire to continually improve their skills through ongoing training in this area, are most likely to translate these beliefs into practical action.

The factor that most strongly predicts positive attitudes towards inclusion is the recognition of the need for training. Expressing a need for training is a significant predictor of positive attitudes, beliefs, and behaviors. This suggests that educators who engage in reflective self-assessment, acknowledge their limitations, and actively seek further training are more receptive to inclusion and more likely to put it into practice. Completing both initial and continuous training courses enhances positive attitudes and beliefs.

Surprisingly, teachers in mainstream schools who do not have students with disabilities in their classrooms exhibit greater openness to inclusion. On the other hand, teaching in special education and within an urban environment predicts less favorable attitudes than mainstream education and rural settings. It appears that direct experience in teaching students with disabilities has a contrasting effect to what is described in the scientific literature, as those who have not had this experience tend to exhibit more positive beliefs regarding inclusion. Notably, certain factors frequently discussed in the literature, such as teaching experience and experience with students with disabilities, do not significantly predict positive inclusive attitudes in the Romanian context.

These findings demonstrate that classroom experiences have not yielded positive outcomes in Romania. To transform these factors into positive ones, the provision of a support system within the classroom, facilitated by the educational system, the school, and specialists, becomes imperative. Also, better tailored and more specific training offers should be considered by the training providers, so that the teachers are better prepared to ensure IE, as they themselves point out.

For future studies, exploring additional factors such as support services provided for teachers, collaboration in multidisciplinary teams, or parental involvement would further enrich our understanding of inclusive attitudes.

**Author Contributions:** Conceptualization, A.M.J., D.B., S.S., A.L. and C.V.B.; methodology, A.M.J., A.L., S.S. and D.B.; formal analysis, A.M.J.; investigation, A.M.J., D.B., C.V.B. and S.S.; data curation, D.B., A.L. and A.M.J.; writing—original draft preparation, A.M.J. and D.B.; writing—review and editing, A.M.J., D.B., S.S., A.L. and C.V.B.; supervision, S.S., A.L. and C.V.B. All authors have read and agreed to the published version of the manuscript.

**Funding:** This research was conducted as part of a project awarded through the grant competition for young researchers, funded by the Faculty of Sociology and Psychology, West University of Timișoara.

**Institutional Review Board Statement:** This study was approved by the Scientific Council of University Research and Creation of West University of Timisoara, approval number 86938.

**Informed Consent Statement:** Informed consent was obtained from all subjects involved in the study.

**Acknowledgments:** We would like to thank the Faculty of Sociology and Psychology of the West University of Timisoara for the grant offered in the framework of the research project for young researchers, in carrying out this research. Thanks to Paul Sârbescu, from West University of Timisoara, for his suggestions on data analysis.

**Conflicts of Interest:** The authors declare no conflict of interest. The funders had no role in the design of the study; in the collection, analyses, or interpretation of data; in the writing of the manuscript; or in the decision to publish the results.

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
