# Peer review of "Exploring Attitudinal Dimensions of Inclusive Education: Predictive Factors among Romanian Teachers"

_education, doi:10.3390/educsci13121224_

Round 1
Reviewer 1 Report
Comments and Suggestions for Authors
Thank you for the opportunity to review your paper. You are delving into a well-researched area, however this study does have the potential to contribute to body of evidence on teacher attitudes to inclusive education. Please see below for some feedback.
Introduction:
The introduction provides a discussion of some literature in this area, but it does not delve as deeply into IE as it could to establish a need for this current study. For example, what is the state of IE globally? What have been some of the successes, some of the challenges? What is the context of schooling and IE in Romania? It is also not clear how IE is understood in this study – what definition did it adhere to?
I wonder if this could be split into different sections:
- Literature
- Measures
- Context
Some further points:
· Avoid the use of the term ‘integration’ when talking about inclusion, as in English they mean different things are represent two different eras of schooling for students with diverse education needs.
· I wonder about the use of the term ‘training’ here. Does a deep understanding of IE – the philosophical, ethical, legal and practical perspectives – require more than training? How does the training happen? Perhaps reconsider this.
· Line 42 – ‘findings on the factors’ of what?
· ‘special education teachers have more favorable attitudes toward IE than do 60 regular education teachers’ – I wonder about the context of these findings feel some further discussion is warranted here.
· ‘Numerous factors were taken into account’ – by who, and for what? This is not clear.
· ‘The affective dimension was put in relation, especially to the previous experience with people with disabilities’ – meaning here is not clear.
Method:
· Table 2 title needs to be reworded for clarity.
· Table 3 title needs to be reworded for clarity.
· Table 4 title needs to be reworded for clarity.
· Table 5 title needs to be reworded for clarity.
· ‘The more initial and in-service training they have attended in special education, the greater their intentions to include, Research hypothesis 3 isn’t supported by the results.’ – reword for clarity.
Discussion:
I would have liked to have the discussion situated more strongly within the Romanian context, as from reading this article I don’t have any understanding about the Romanian context – its unique characteristics and journey to date to become more inclusive. This also needs to be brought into the first section of the article (as noted above).
Some more points:
· ‘Firstly, the results showed that the factor that most significantly predicts teachers' 330 inclusive attitudes is their need for training in special education.’ It is not clear in the article to here that you have been talking about the teachers need for further professional learning, but rather it reads as though you were looking at current levels of education teachers have received. Please review and clarify this.
· You have made links between professional learning in special education and more positive attitudes towards inclusive education. This needs further exploration here and also perhaps clear definitions of what you mean when you are using the terms special education and inclusive education.
· ‘his or her limits’ – keep language gender neutral.
· ‘Also, the problems of inclusion in Romania are directly experienced by teachers in special education, together with the children reoriented from mainstream to special schools, a failure of the inclusive school.’ – this needs further exploration. There is also much work that suggests that special education educators and researchers have presented as a barrier to inclusive education. This needs to be explored. As does what you mean when you say a failure of the inclusive school.
· ‘disabled person’ – use person first language.
· ‘Through real classroom support, provided by the system, school, and specialists. Future studies could investigate the extent to which this support is actually provided and the way this support influences teachers' attitudes.’ – this needs to be expanded upon.
· Limitations could be a separate section.
Conclusion:
· ‘the need for training and the initial training’ – the distinction between these is not clear within this article.
· It would be good to perhaps include a little more about a way forward rather than just rewording the summary provided above.
Comments on the Quality of English LanguageSome tweaking needed (as per feedback), but generally it reads well.
Author Response
Dear reviewer,
Thank you for your valuable review and recommendations! Your comments have been immensely helpful, as we desire to improve the article. We appreciate your time and feedback. The table below illustrates how we addressed the recommendations received:
Reviewer`s comments |
Authors` response |
Introduction: |
|
The introduction provides a discussion of some literature in this area, but it does not delve as deeply into IE as it could to establish a need for this current study. For example, what is the state of IE globally? What have been some of the successes, some of the challenges? What is the context of schooling and IE in Romania? It is also not clear how IE is understood in this study – what definition did it adhere to? |
We deepened the Ro context, a distinct paragraph was added in response to the reviewers' comments (lines 28-32 and 39-52). It was not the focus of the article to deal with the successes of IE globally and in Romania, and we were not sure how to integrate these aspects to fit into the article focus. |
I wonder if this could be split into different sections: - Literature - Measures - Context |
We made distinct paragraphs in the Literature part with the Context (lines 53-58), and the state of investigation of the teachers attitude (lines 119-127).
|
Some further points: · Avoid the use of the term ‘integration’ when talking about inclusion, as in English they mean different things are represent two different eras of schooling for students with diverse education needs. · I wonder about the use of the term ‘training’ here. Does a deep understanding of IE – the philosophical, ethical, legal and practical perspectives – require more than training? How does the training happen? Perhaps reconsider this. · Line 42 – ‘findings on the factors’ of what? · ‘special education teachers have more favorable attitudes toward IE than do 60 regular education teachers’ – I wonder about the context of these findings feel some further discussion is warranted here. · ‘Numerous factors were taken into account’ – by who, and for what? This is not clear. · ‘The affective dimension was put in relation, especially to the previous experience with people with disabilities’ – meaning here is not clear. |
All terms related to integration were replaced with inclusion terminology. A paragraph was added in response to the reviewers' comments about the training ( lines 70-74) The sentence in line 42 was corrected (lines 65-67). A reference to special education teachers` attitudes was added in response to the reviewers' comments (lines 117-118). The sentence about the affective dimension was rephrased (lines 131-132). |
Method: |
|
· Table 2 title needs to be reworded for clarity. · Table 3 title needs to be reworded for clarity. · Table 4 title needs to be reworded for clarity. · Table 5 title needs to be reworded for clarity. · ‘The more initial and in-service training they have attended in special education, the greater their intentions to include, Research hypothesis 3 isn’t supported by the results.’ – reword for clarity. |
The table titles have been shortened and revised. (lines 264-265, 300-301, 334-335, 365). The paragraph was rephrased (lines 350-353). |
Discussion: |
|
I would have liked to have the discussion situated more strongly within the Romanian context, as from reading this article I don’t have any understanding about the Romanian context – its unique characteristics and journey to date to become more inclusive. This also needs to be brought into the first section of the article (as noted above). |
In the discussion, a paragraph about barriers in Romania to inclusive education was added (lines 389-390). In the introduction part, a paragraph about the journey of inclusive education in Romania was added (lines 39-52). |
Some more points: · ‘Firstly, the results showed that the factor that most significantly predicts teachers' 330 inclusive attitudes is their need for training in special education.’ It is not clear in the article to here that you have been talking about the teachers need for further professional learning, but rather it reads as though you were looking at current levels of education teachers have received. Please review and clarify this. · You have made links between professional learning in special education and more positive attitudes towards inclusive education. This needs further exploration here and also perhaps clear definitions of what you mean when you are using the terms special education and inclusive education. · ‘his or her limits’ – keep language gender neutral. · ‘Also, the problems of inclusion in Romania are directly experienced by teachers in special education, together with the children reoriented from mainstream to special schools, a failure of the inclusive school.’ – this needs further exploration. There is also much work that suggests that special education educators and researchers have presented as a barrier to inclusive education. This needs to be explored. As does what you mean when you say a failure of the inclusive school. · ‘disabled person’ – use person first language. · ‘Through real classroom support, provided by the system, school, and specialists. Future studies could investigate the extent to which this support is actually provided and the way this support influences teachers' attitudes.’ – this needs to be expanded upon. · Limitations could be a separate section.
|
The need for the training concept is explained in more detail at lines 398-400.
Literature reviews demonstrating the connection between professional training and teachers' inclusive attitudes were added (lines 117-118).
The distinction between the terms was clarified and added in the theoretic part (lines 28-32)
We revised the language to be gender neutral (line 417)
The sentences were rephrased (lines 433-435).
The term "disabled person" has been replaced throughout the document with "person with a disability."
A separate limitation section was created (lines 593-507) |
Conclusion: |
|
‘the need for training and the initial training’ – the distinction between these is not clear within this article |
Distinction between need for training and initial training was explained at lines 524 - 529. |
It would be good to perhaps include a little more about a way forward rather than just rewording the summary provided above. |
We revised accordingly in lines 552-554. |
Many thanks for your recommendations and appreciation. We hope the manner in which we have addressed them is convincing.
With gratitude, the authors
Reviewer 2 Report
Comments and Suggestions for Authors
Abstract
Suggestions for Improvement:
· Consider providing more specific details regarding the methodology employed (e.g., sampling technique, data collection tools) for a clearer understanding of the research approach.
· Elaborate on the nuanced cultural implications of the identified need for training in special education within the context of Romanian culture.
· Specify the nature and extent of the predictive relationship between the identified factors and inclusive attitudes to strengthen the discussion of findings.
· Incorporate a concise outline of the limitations encountered during the research process for a more comprehensive understanding of potential constraints.
Introduction
The introduction provides a comprehensive overview of inclusive education's evolution, specifically in Romania, emphasizing its transformation from segregated schooling to gradual integration. It highlights the global shift towards inclusive practices and the importance of teachers' attitudes in implementing inclusive education. The study's objective to explore predictive factors influencing teachers' attitudes towards inclusive education within the Romanian cultural context is clearly articulated.
Strengths:
· Provides historical context: The introduction effectively traces the historical progression of inclusive education, particularly within Romania, offering insight into the country's transition from segregated to integrated schooling.
· Clear research objective: The introduction precisely outlines the study's aim of identifying predictive factors impacting teachers' attitudes toward inclusive education in Romania.
· References to existing literature: It integrates previous research findings, contributing to a contextualized understanding of the topic.
· Emphasizes the significance of teachers' attitudes: Recognizes the substantial influence of teachers' attitudes on implementing inclusive practices.
Suggestions for Improvement:
· Clarify research gap: While the introduction mentions the need for further studies in less developed countries to generalize findings on factors impacting attitudes, it would benefit from a more explicit statement on the specific gap this study aims to fill within the literature.
· Streamline information: Some sections provide extensive lists of factors without a clear transition, which might be overwhelming for readers. Consider organizing the information more concisely for better readability.
· Methodological context: Introduce or briefly outline the methodological approach (e.g., research design, data collection methods) to provide readers with an understanding of how the study aims to address the identified research gap.
Methodology
The methodology employed a quantitative, non-experimental, correlational research design to investigate factors predicting teachers' attitudes toward inclusive education in Romania across cognitive, affective, and behavioral dimensions. The study utilized the Multidimensional Attitudes towards Inclusive Education Scale (MATIES) to measure these dimensions, demonstrating good internal consistency within the scale. The study's design and the instrument chosen are appropriate for addressing the research question.
Strengths:
· Clear research design: The research design suits the study's objective of identifying predictive factors influencing teachers' attitudes toward inclusive education.
· Utilization of established measurement tool: The use of the MATIES scale, known for its psychometric properties and focus, is a strength as it offers a standardized approach to measuring attitudes.
· Comprehensive participant information: The detailed demographic information about participants provides a comprehensive understanding of the sample characteristics, allowing potential insights into factors influencing attitudes.
Suggestions for Improvement:
· Justification of hypotheses: While the study presents hypotheses predicting factors for each attitudinal dimension, it might benefit from further explanation or empirical reasoning behind these hypotheses to enhance the understanding of the expected associations.
· Clarification on data collection: The description of data collection is informative but lacks information on response rates or methods used to ensure data quality and reliability. Including details about response rates and any steps taken to ensure the survey's validity and reliability would strengthen the methodology section.
· Further elaboration on instrument adaptation: While the retroverse translation technique was mentioned for adapting the instrument to the Romanian context, more details on this process, including potential challenges faced or steps taken to ensure cultural appropriateness, would be beneficial.
Results:
This section presents findings regarding factors predicting teachers' attitudes towards inclusive education across cognitive, affective, and behavioral dimensions. The data analysis uses multiple regression models to assess the impact of various factors. Here's a critical evaluation:
Strengths:
· Clear presentation: The results are systematically organized, using tables to illustrate the regression coefficients, making it easy to follow and interpret.
· Statistical significance: The analysis rigorously examines the statistical significance of each predictor variable for each dimension, enhancing the study's credibility.
· Consistent reporting: The section consistently reports the significance levels (p-values), regression coefficients, confidence intervals, and standardized coefficients for each predictor, ensuring comprehensive reporting of statistical outcomes.
Suggestions for Improvement:
· Clarity in interpretation: While statistical significance is reported, the interpretation and practical implications of these findings could be expanded. Explaining the real-world implications of these statistical associations would add depth to the analysis.
· Coefficients explanation: Providing more context on the interpretation of coefficients would enhance understanding. For instance, explaining how a unit change in each predictor affects the respective attitude dimension would make the findings more accessible.
· Supporting evidence for hypotheses: Although the results are presented, providing additional context or referencing theoretical frameworks that support or contradict the hypotheses could enrich the analysis.
· Contextual insights: The section lacks qualitative insights or explanations regarding why certain factors might have a stronger influence on attitudes. Incorporating teacher perspectives or contextual information could enrich the interpretation.
Discussion and Conclusions:
This section discusses and draws conclusions from the study's findings, offering insights into factors influencing teachers' attitudes toward inclusive education in the Romanian context. Here's a critical evaluation:
Strengths:
· Contextualized discussion: The discussion offers a comprehensive analysis of the factors influencing inclusive attitudes, contextualizing findings within the Romanian educational system. It effectively references prior studies to support and contrast its results, adding depth to the discussion.
· Integration of findings: The section synthesizes results across attitudinal dimensions (cognitive, affective, and behavioral), providing a holistic understanding of how these dimensions interplay with various predictors.
· Identification of implications: It identifies practical implications, such as the need for tailored training programs in special education, addressing the expressed needs of teachers, and suggests potential avenues for future research.
Suggestions for Improvement:
· Clarity in interpretation: While the discussion presents various factors influencing attitudes, it could further clarify the complex relationships between predictors and attitudes. Providing additional context or possible reasons for unexpected findings would enhance the interpretation.
· Addressing limitations: The section acknowledges some limitations, such as the lack of data on the degree of disability and unexplored factors. Expanding on these limitations and discussing their potential impacts on the findings could strengthen the study's credibility.
· Integration of theoretical frameworks: While referencing prior studies, integrating relevant theoretical frameworks or models that could explain or support the findings might add theoretical depth to the discussion.
Author Response
Dear reviewer,
Thank you for your valuable review and recommendations! Your comments have been immensely helpful, as we desire to improve the article. We appreciate your time and feedback. The table below illustrates how we addressed the recommendations received:
Reviewer`s comments |
Authors` response |
Abstract |
|
Consider providing more specific details regarding the methodology employed (e.g., sampling technique, data collection tools) for a clearer understanding of the research approach. |
A reference to the participant selection method was added in response to the reviewers' comments (line 11). The data collection tools are specified in line 12. |
Elaborate on the nuanced cultural implications of the identified need for training in special education within the context of Romanian culture. |
We appreciate the reviewer's insightful comment regarding the nuanced cultural implications of the identified need for training in special education within the context of Romanian culture. Given the constraints of the abstract length (200 words), a detailed exploration of these cultural implications could be found in the discussion section. |
Specify the nature and extent of the predictive relationship between the identified factors and inclusive attitudes to strengthen the discussion of findings. |
A reference to the factors` predictive power was added in response to the reviewers' comments (lines 16-18). |
Incorporate a concise outline of the limitations encountered during the research process for a more comprehensive understanding of potential constraints |
A reference to the most important limitation was added in response to the reviewers' comments (lines 16-18). |
Introduction
|
|
Clarify research gap: While the introduction mentions the need for further studies in less developed countries to generalize findings on factors impacting attitudes, it would benefit from a more explicit statement on the specific gap this study aims to fill within the literature.
|
A more detailed explanation of the research gap was added in response to the reviewers' comments (lines 81-87). |
Streamline information: Some sections provide extensive lists of factors without a clear transition, which might be overwhelming for readers. Consider organizing the information more concisely for better readability.
|
To enhance readability, the paragraph was rephrased (lines 119-127). |
Methodological context: Introduce or briefly outline the methodological approach (e.g., research design, data collection methods) to provide readers with an understanding of how the study aims to address the identified research gap. |
A brief description of the methodological approach was added (lines 87-90)
|
Methodology |
|
Justification of hypotheses: While the study presents hypotheses predicting factors for each attitudinal dimension, it might benefit from further explanation or empirical reasoning behind these hypotheses to enhance the understanding of the expected associations. |
Explanations for the hypotheses were added (lines 165-167, 171-173, 177-178, 181-186) |
Clarification on data collection: The description of data collection is informative but lacks information on response rates or methods used to ensure data quality and reliability. Including details about response rates and any steps taken to ensure the survey's validity and reliability would strengthen the methodology section. |
Clarification on data collection was added (lines 194-196). |
Further elaboration on instrument adaptation: While the retroverse translation technique was mentioned for adapting the instrument to the Romanian context, more details on this process, including potential challenges faced or steps taken to ensure cultural appropriateness, would be beneficial. |
A reference to the instrument translation was added in response to the reviewers' comments (lines 222-223). |
Results |
|
Clarity in interpretation: While statistical significance is reported, the interpretation and practical implications of these findings could be expanded. Explaining the real-world implications of these statistical associations would add depth to the analysis. |
To clarify the results, a short sentence was added to each of them, keeping in mind the need to be brief (lines 283-287; 314-319; 350-352; 380-384) |
Coefficients explanation: Providing more context on the interpretation of coefficients would enhance understanding. For instance, explaining how a unit change in each predictor affects the respective attitude dimension would make the findings more accessible. |
The statistical coefficients were named in the note (legend) below each table. An interpretation of each result was added (lines 283-287; 314-319; 350-352; 380-384) |
Supporting evidence for hypotheses: Although the results are presented, providing additional context or referencing theoretical frameworks that support or contradict the hypotheses could enrich the analysis. |
We have added short comments to support evidence for the hypotheses (lines 287-289, 318-319, 3551-352). |
Contextual insights: The section lacks qualitative insights or explanations regarding why certain factors might have a stronger influence on attitudes. Incorporating teacher perspectives or contextual information could enrich the interpretation. |
To clarify the results, a short sentence was added to each of them. (lines 283-287; 314-319; 350-352; 380-384) |
Discussion and conclusion |
|
Clarity in interpretation: While the discussion presents various factors influencing attitudes, it could further clarify the complex relationships between predictors and attitudes. Providing additional context or possible reasons for unexpected findings would enhance the interpretation. Contextualized discussion: The discussion offers a comprehensive analysis of the factors influencing inclusive attitudes, contextualizing findings within the Romanian educational system. It effectively references prior studies to support and contrast its results, adding depth to the discussion. |
The relationship between the first predictor, the teachers` need for training, and their attitudes is explained in lines 401-404, The relationship between the second predictor, the school type, and their attitudes is explained in a sentence added in lines 429-430. The relationship between the third predictor, the relationship with a person with disabilities and their attitudes is explained in a sentence added in lines 451-452. |
Addressing limitations: The section acknowledges some limitations, such as the lack of data on the degree of disability and unexplored factors. Expanding on these limitations and discussing their potential impacts on the findings could strengthen the study's credibility. |
Another limitation was added in lines 501-507 |
Integration of theoretical frameworks: While referencing prior studies, integrating relevant theoretical frameworks or models that could explain or support the findings might add theoretical depth to the discussion. |
The results are interpreted in the discussion section in relation to the Knowledge - Attitude - Behavior (KAB) theory, and more theoretic considerations in this line are added (lines 422-426). The results are interpreted in the discussion section in relation to the Theory of Planned Behavior and more theoretic considerations in this line are added (lines 442-448).
|
Many thanks for your recommendations and appreciation. We hope the manner in which we have addressed them is convincing.
With gratitude, the authors
Reviewer 3 Report
Comments and Suggestions for Authors
Thank you for submitting your manuscript for consideration. Your paper presents a fascinating exploration into the realm of inclusive education, particularly within the Romanian context. The methodology and statistical analyses employed are commendable for their thoroughness and clarity, effectively addressing each dimension of the study. Additionally, the choice of topic and research tools demonstrates a well-considered approach to investigating this significant area of education.
However, there are a few areas where the paper could be enhanced:
Comparative Analysis with Other Post-Soviet Countries: The paper would benefit greatly from a comparison with other post-Soviet countries such as Poland, Slovakia, Hungary, and the Czech Republic. This comparison could provide valuable context and deepen the understanding of Romania's unique position in terms of inclusive education practices and attitudes. It would be particularly insightful to explore how Romania's post-Soviet experience has shaped its approach to inclusive education and how it compares with its neighbors who share a similar historical background.
Exploration of Theoretical Frameworks: The paper touches upon the Theory of Planned Behaviour but does not delve into why this theory does not seem to align with the experiences of teachers who have worked with children with disabilities in Romania. It would be beneficial to discuss other relevant theories, such as the Knowledge-Attitude-Behavior (KAB) theory and the Theory of Reasoned Action, to provide a more comprehensive theoretical framework. This exploration could help explain the unique findings in your study and provide a richer theoretical context.
Clarification and Expansion of Theoretical Discussion: I recommend expanding the discussion section to address why certain theoretical expectations (such as those posited by the Theory of Planned Behaviour) are not met in the context of Romanian teachers with experience in teaching children with disabilities. This could include an exploration of cultural, systemic, or educational factors that might influence these outcomes.
In summary, while the paper is well-constructed and offers significant insights into inclusive education in Romania, incorporating a comparative analysis with other post-Soviet countries and a deeper exploration of various theoretical frameworks could greatly enhance its contribution to the field. These additions would not only enrich the discussion but also offer a more nuanced understanding of the complex dynamics at play in inclusive education within post-Soviet contexts.
I look forward to seeing the revised version of your manuscript.
Author Response
Dear reviewer,
Thank you for your valuable review and recommendations! Your comments have been immensely helpful, as we desire to improve the article. We appreciate your time and feedback. The table below illustrates how we addressed the recommendations received:
Comparative Analysis with Other Post-Soviet Countries: The paper would benefit greatly from a comparison with other post-Soviet countries such as Poland, Slovakia, Hungary, and the Czech Republic. This comparison could provide valuable context and deepen the understanding of Romania's unique position in terms of inclusive education practices and attitudes. It would be particularly insightful to explore how Romania's post-Soviet experience has shaped its approach to inclusive education and how it compares with its neighbors who share a similar historical background.
|
A reference to the Post- communist countries need for teacher training was added in response to the reviewers' comments (lines 406-410) A reference to the Post- communist countries inadequate classroom support was added in response to the reviewers' comments (lines 472-474).
|
Exploration of Theoretical Frameworks: The paper touches upon the Theory of Planned Behaviour but does not delve into why this theory does not seem to align with the experiences of teachers who have worked with children with disabilities in Romania. It would be beneficial to discuss other relevant theories, such as the Knowledge-Attitude-Behavior (KAB) theory and the Theory of Reasoned Action, to provide a more comprehensive theoretical framework. This exploration could help explain the unique findings in your study and provide a richer theoretical context.
|
A reference to Knowledge-Attitude-Behavior (KAB) theory was added (lines 101- 107) A reference to the Theory of Planned Behavior was added (lines 107- 113). |
Clarification and Expansion of Theoretical Discussion: I recommend expanding the discussion section to address why certain theoretical expectations (such as those posited by the Theory of Planned Behaviour) are not met in the context of Romanian teachers with experience in teaching children with disabilities. This could include an exploration of cultural, systemic, or educational factors that might influence these outcomes. |
The results were interpreted in the discussion section in relation to the Knowledge - Attitude - Behavior (KAB) theory (lines 422-426). The results were interpreted in the discussion section in relation to the Theory of Planned Behavior (lines 442-448). |
Many thanks for your recommendations and appreciation. We hope the manner in which we have addressed them is convincing.
With gratitude, the authors
Round 2
Reviewer 1 Report
Comments and Suggestions for Authors
Thank you for your responses to the feedback, and for taking the time to respond to it. The article has certainly been strengthened by your diligence in responding to the feedback of all three reviewers. I would suggest that it requires a small amount of editing from here, just to tighten the writing, particularly around the areas where text has been added - to make sure the paragraphs structure works and the sentences flow. Once this has been done, the paper will be ready to go. Insight into these less well researched contexts are valuable, and I look forward to seeing more of your work in the future.
Comments on the Quality of English LanguageI would suggest this paper requires a small amount of editing from here, just to tighten the writing, particularly around the areas where text has been added - to make sure the paragraphs structure works and the sentences flow. Once this has been done, the paper will be ready to go.
Author Response
Dear reviewer,
Thank you for your invaluable review and recommendations! Your comments have proven to be exceptionally helpful in enhancing the article.
Reviewer`s comments |
Authors` response |
I would uggest that it requires a small amount of editing from here, just to tighten the writing, particularly around the areas where text has been added - to make sure the paragraphs structure works and the sentences flow. Once this has been done, the paper will be ready to go. Insight into these less well researched contexts are valuable, and I look forward to seeing more of your work in the future. |
For improved fluency and coherence of ideas and discourse, the following modifications were implemented: The paragraphs pertaining to the history of inclusive education in Romania have been relocated to a distinct subsection, titled 1.1 The Context of Inclusive Education in Romania (lines 66-92) The paragraph concerning initial and continuous training has been relocated to lines 47-52. The subtitle "Teachers' Attitudes towards Inclusive Education" was assigned the number 1.2 and is located at line 93. Lines 430-431 were rephrased. A reference to the degree of disability was incorporated in lines 439-440. The second study limitation, initially found in the discussion section, has been relocated to the limitation section, now residing in lines 495-501. The bibliographic references have been renumbered to align with the revised structure of the text. |
|
|